# A case study of the features and holistic athlete impacts of a UK sports-friendly school: Student-athlete, coach and teacher perspectives

Ffion Thompson[1]*, Fieke Rongen[1], Ian Cowburn[1], Kevin Till[1,2]

1 Carnegie School of Sport, Leeds Beckett University, Leeds, United Kingdom, 2 Leeds Rhinos Rugby League Club, Leeds, United Kingdom

* ffion.thompson@leedsbeckett.ac.uk

**Data Availability Statement:** All relevant data are within the paper and its Supporting Information file.

## Abstract

In order to understand the features of sport schools and their impacts on the holistic development of student-athletes, it is important to take into account the voice of multiple stakeholders central to the programmes (student-athletes, coaches, teachers). Through a case-study approach, using five focus groups, with 19 student-athletes, and six semi-structured interviews with three coaches and three dual coach and teachers, this study explored the perceived impacts of one sport-friendly school (pseudonym–"Salkeld High") on holistic athlete development and the features that drove these impacts. Using a critical realist approach to thematic analysis, findings indicated a multitude of immediate, intermediate and long-term positive and negative impacts associated with academic/vocational (e.g., academic security vs. second/third choice university), athletic/physical (e.g., performance development vs. injuries), psychosocial (e.g., social skills vs. social scarifies) and psychological (e.g., sport confidence vs. performance pressure) development of "Salkeld High" student-athletes. Overall, "Salkeld High" was viewed as an integrated school environment for sport, academics, and boarding, where academic (e.g., extra-tutoring), athletic (e.g., high volume/frequency of training), and psychosocial/psychological (e.g., pastoral services) features are all in one location. The student-athletes tended to get a well-rounded, balanced holistic experience. However, the intensified and challenging nature of involvement did present some negative impacts that stakeholders should be aware of when designing, implementing, and evaluating sport-friendly school programmes. Furthermore, although "Salkeld High" was seen as an integrated environment within the school, it could do better at collaborating with wider sporting structures.

## 1 Introduction

Over the last two decades, significant emphasis has been placed on the development of youth sport, increasing the global distribution of youth athletic and talent development programmes

**Funding:** The author(s) received no specific funding for this work.

**Competing interests:** The authors have declared that no competing interests exist.

[1]. The International Olympic Committee recommends that youth athletic development programmes, including Talent Identification and Development Systems (TIDS), focus on developing healthy, capable and resilient young athletes while achieving widespread, inclusive, sustainable and enjoyable participation and success [2]. Yet, due to the increased intensity and professionalisation of youth sport programmes achieving this aim may pose a considerable challenge for all stakeholders (e.g., coaches, sport governing bodies). While research has emphasised the positive impacts (e.g., enhanced physiological capacity, increased confidence, high academic achievers) of youth sports programmes (e.g., [3]), recent position and consensus statements [2,4] have also warned of the risks (e.g., injury, performance sports focus, educational sacrifice). As such, given that most youth athletes do not ultimately succeed in their sport, ensuring the healthiness of sport involvement is paramount, but may require a considerable balancing act [3,4].

Worldwide, the term 'dual career' has been introduced to help aid the specific challenges elite youth athletes face in balancing sport and education [5]. Recent research (e.g., [6–8]) has demonstrated that the combination of sport and education has multiple benefits, including a balanced lifestyle, increased well-being, life skill development, self-regulation abilities, and expanded social support networks [5]. However, challenges associated with dual career pursuits have also been reported, including increased demands, high-stress levels, and the need for additional support. Therefore, a holistic whole-person approach to an athlete's development has been promoted for overcoming the challenges of dual careers [9] and is considered an essential feature of successful talent development environments [10].

In an attempt to pursue such a whole-person approach, researchers have increasingly followed Wylleman's [11] Holistic Athletic Career model, where for healthy development, youth sports programmes should embrace the multi-dimensional nature of youth athlete development. Although practitioners may instinctively focus on assessing and monitoring physical and sports performance measures, for the holistic development of youth athletes, consideration must be given to the psychosocial, psychological and academic/vocational domains [11]. Furthermore, research demonstrates the strong concurrent, interactive and reciprocal nature of transitions occurring in the sporting career (athletic transitions) and in other domains (e.g., academic, psycho-social, professional) [11,12], as well as the complex and dynamic nature of dual career environments [10]. One example of a dual career development environment that aims to cater for youth athletes' holistic development is a sports school. Csikszentmihalyi et al. [13] argue that schools are valuable resources due to young athletes' access to coaches and teachers who create a positive environment for development. Furthermore, research suggests that specialised sport schools have made life easier for many athletes [14,15]. Recently, Morris et al. [16] categorised two types of sport schools; sports-friendly schools and elite sports schools. A sports-friendly school is defined as a *"regional educational institution, who permit elite sport or align themselves with elite sport to provide academic flexibility for athletes to train and compete in their own sporting environment"* ([16], p.140). An elite sports school is defined as an *"educational institution purposefully developed for elite athletes who wish to combine their athletic and academic careers"* ([16], p.140). Both sports-friendly schools and elite sports schools are situated in lower and upper general and vocational secondary education (International Standard Classification of Education level 2–5). However, unlike a sports-friendly schools, an elite sports school has formal communication with a sports federation, often receiving funding.

Overall, sport schools offer student-athletes considerable academic support (e.g., adaptation of school and training schedules, lighter load by one subject) and athletic support (e.g., high-quality coaches, physiotherapy; [17,18]). Whilst a non-sport school may offer time off to practice and might adapt the school day, they usually cannot offer the same range of support

services as a sport school due to resources available (e.g., finances; [19]). Likewise, sport schools may also have an advantage over club-based sport development as they can better manage the student-athletes competing demands (e.g., time) due to resource efficiency (e.g., limited travel time extra support services; [14]). Rearranging exam dates due to competitions, part-time study, and a holistic club culture that is supportive of life outside of sport has been shown to enable student-athletes to manage both sport and academics [20,21]. Therefore, a sport school (with the effective combination of competitive sports, education, and accommodation) could favour future top sporting performances while safeguarding education priorities [22], alongside potentially allowing for more 'free time' through optimised time schedules. However, due to the considerable challenge of balancing sport, academics, and other vital aspects of a student-athlete's life (e.g., social life), sport school student-athletes may still suffer from some of the same challenges of intensified youth sport.

Indeed, many immediate, intermediate and long-term positive and negative impacts have been associated with being a sport school student-athlete [23]. Potential impacts include physical (e.g., increased physical fitness vs injury), psychological (e.g., self-optimisation vs performance pressure), psychosocial (e.g., life skills vs time away from family), and educational (e.g., academic success vs limited experience with ordinary life outside of competitive sport; [23]). Yet, several limitations exist within the current evidence base, including; 1) the range of sport school features have not been evaluated in-depth specifically in sports-friendly schools (e.g., athletic and academic support services); 2) limited research examines how sports-friendly school features are operationalised in different contexts (e.g., countries); 3) current research fails to provide multi-dimensional evaluations of athlete impact, often focussing on one or two dimensions; 4) limited research evaluates how features affect athlete impacts (i.e., it is not possible within the current literature to establish a rigorous causal relationship between the characteristics and features of sports school and holistic athlete impacts), and 5) limited studies take into account the voice of multiple stakeholders within these programmes. Furthermore, while there is research that has provided a basic understanding of the overall features and holistic impacts, due to the varying approaches to dual career support [17,24], it is also important to understand environments within a particular context.

Overall, within the UK, there are substantially more sports-friendly schools, with only one identified example of an elite sports school found in Scotland [25]. Sports-friendly schools in the UK tend to be more independent than the systemic approach in other countries (e.g., Germany, Sweden; [26]). In the UK, the development of a sports-friendly school is primarily a matter for individual schools and is often pursued as part of a strategy to create a distinct identity. Therefore, sports programmes are likely to differ depending upon the context, and each sports-friendly school may have slightly different aims/aspirations. As a result, it is important to investigate the individual contexts of sports-friendly schools within the UK and the stakeholders' views at the centre of these programmes, namely, the youth athletes (centre of the programme) and coaches/teachers (those that drive the programmes).

Based on the above, this study aimed to identify, explore and understand the features of a sport friendly school in the UK and their impact on the holistic development of student-athletes, from the perspective of the student-athletes, coaches and teachers.

## 2 Methods

### 2.1 Research approach

This study adopted a critical realist (CR) approach. Critical realism is appropriate for determining causal explanations [27], where environments generate impacts/outcomes through multiple interacting features [28]. The CR search for causation helps researchers explain social

events and suggest practical policy recommendations to address social problems. Critical realists are concerned with understanding how particular impacts are generated by the interaction between features and contexts [29]. As such, a qualitative research methodology was adopted [30] as this enabled the researcher to evaluate what student-athletes and staff perceived to be the impacts of student-athlete sports-friendly school involvement and how these impacts were brought about by the features of the school and how they differed across individuals [31]. Using semi-structured interviews and focus groups, the primary researcher was not constrained by the questions but guided by them [32]. As a result, interviewees had the freedom to discuss personally important issues [33]. Employing semi-structured interviews and focus groups is compatible with a CR approach, which advocates that there must remain a degree of flexibility in how interview questions are designed to elicit information about the unique interaction of causal features in a particular context [34] and to enable an in-depth, contextualised understanding [35,36]. Furthermore, the complex and dynamic nature of sport-friendly school environments and the objective of developing a rich and detailed insight into actual existing environments called for a case study approach [10]. A case study approach is the most common, and arguable one of the most useful, form of CR research [37]. As such, this study deliberately focused on one case study to generate an in-depth, multi-faceted exploration of a complex environment in its real-life setting [38].

## 2.2 Context of study

One sports-friendly school (pseudonym–"Salkeld High") was selected as the case for the study based on the Morris et al. [16] definition of a sports-friendly school. The selection of "Salkeld High" was information-oriented and opportunistic. "Salkeld High" was an established and mature environment, with seven years of experience providing dual career support through a performance sport pathway embedded within a UK independent school. The school has eight performance sports as part of its performance programmes: athletics, basketball, cricket, football, hockey, netball, rugby, and swimming, targeted at year groups 7–13 (aged 12–18 years). Student-athletes accepted onto the performance sport program at "Salkeld High" pay for or receive (via scholarship) a place to study, live, and train during their secondary school years. Based on the information above, we envisioned that the stakeholders (i.e., staff and student-athletes) embedded within "Salkeld High" would offer information-rich experiences widening the understanding holistic impacts of sports-friendly schools and how these may be connected to the school's features.

## 2.3 Positionality of the researcher

Three days a week, the primary researcher (first author) works within "Salkeld High" as the lead strength and conditioning (S&C) coach and is embedded within the context. Furthermore, the primary researcher was previously a competitive student-athlete at a different sports-friendly school. As such, the primary researcher had a unique understanding of sports-friendly schools, including boarding, sport and education demands (dual career development), alongside the support services offered and the multiple interactions student-athletes have with other stakeholders within an of sports-friendly school performance programme. The first-hand experiences of sports-friendly school environments allowed the researcher to have a contextually relevant understanding of the various stakeholders' language, behaviours, and viewpoints, resulting in a more-depth understanding for conducting the study on holistic athlete impacts and underlying features of of sports-friendly school programmes [39].

## 2.4 Participants

Nineteen student-athletes (ten men and nine women, aged between 15–18, mean age = 16.5 ± 0.7 years) and six staff (one woman and five men, aged between 41–64, mean age = 47.3 ± 9.0 years) participated in the study. Participants had to meet one of the following criteria: (a) be a staff member at "Salkeld High" or, (b) participate as an athlete on "Salkeld High's" performance sport program. Overall, out of the 19 student-athletes, nine were boarders, and ten were non-boarders, representing the following sports: hockey (n = 4), netball (n = 4), football (n = 4), rugby (n = 3) and basketball (n = 4). Hockey and netball student-athletes were all women, basketball and rugby were all men, and football had three men and one woman representative. The student-athletes had been attending and competing at "Salkeld High" for an average of 2.5 years and had first-hand practical experience of being embedded within a sports-friendly school performance sports programme and the corresponding integrated educational institute. As such, the student-athletes were purposefully selected based on their understanding of being a student-athlete within "Salkeld High".

Overall, the staff were purposefully sampled based upon coaching experience within sports-friendly schools, comprehensive knowledge of "Salkeld High" system, varied perspectives across five sports, regular contact with the student-athletes, and multiple communication links with other staff within "Salkeld High." The staff had worked in coaching for between 11 and 42 years (mean coaching experience = 21.5 ± 11.6 years), and at "Salkeld High" for an average of 5.3 ± 3.3 years, and all had experience in more than one sports-friendly school. Furthermore, three out of the six staff members held a dual role at the school (i.e., were both a coach and a teacher). As such, the staff provide a varied perspective on holistic outcomes and a more comprehensive viewpoint on how the performance sports programme and integrated educational institute functioned.

## 2.5 Data collection

This study was granted by university sub-ethics committee (Ref. 73464). The primary researcher contacted the participants by email and provided them with information about the research, highlighting the risks and benefits of the study. Before the interviews, written informed consent was received from all participants and parents for those under the age of 16 (n = 1) received a parent information letter with a parental opt out form also provided (which was not signed). Six 30–60 minute (34.4 ± 12.5 minutes) semi-structured interviews with staff and five 30–60-minute focus groups (42.3 ± 12.7 minutes) with 3–4 student-athletes from the same sport were conducted. Keeping a similar structure, separate interview guides were used for the student-athletes and the staff to allow for different perspectives and match the interviewees' experiences and roles. For example, due to the staff's extensive experience within "Salkeld High", they could reflect more on the "Salkeld High" long-term impacts, where the student-athletes would not have experienced these long-term implications. All participant interview guides were divided into three parts. In the introductory part participants background and immediate impressions of the environment were explored. In the descriptive part, questions focussed on the features of "Salkeld High" programme (e.g., what extra support services are on offer as part of the performance sport pathway?) and the positive and negative holistic athlete impacts of being part of "Salkeld High" (e.g., can you tell me what you believe to be the positive impacts/benefits associated with being a student-athlete in the sports school programme?). In this section, staff had additional questions on long-term impacts, which were not included in the student-athlete guide. Finally, the explanatory part examined the factors contributing to the environment's impacts, which comprised questions about the effect of "Salkeld High" features on these impacts (e.g., how do you think the sports school influenced these

negative impacts?). In order to systematically address what student-athletes and staff perceive to be the positive and negative impacts of a sports-friendly school on holistic athletes' development, the interview guide was developed based on the Holistic Athletic Career model [11] in order to ensure a whole-career and a whole-person perspective of holistic development. The Holistic Athletic Career model has been extensively used in previous dual career studies to guide information collection about the athlete as a whole person (e.g., [8]). As such, based on this model, questions were asked about athletic/physical, academic/vocational, psychosocial and psychological impacts.

## 2.6 Data analysis

In line with our CR stance, thematic analysis (TA) was thought to provide the most useful data analysis framework. It enabled the researchers to interpret the participant's experience, the context of these experiences, and the features that drove the perceived impacts. Overall, the analysis was shaped to reflect existing theory while remaining open to novel ideas that may help to inform theories about why things happen [40].

This study aimed to provide an exploratory (i.e., seeking to describe a phenomena) and an explanatory (i.e., seeking to explain the causes of phenomena) approach to research [41]. As such in analysing the data, Fryers' [42] five-step critical realist approach to TA was aligned with Wiltshire and Ronkainen [43] three types of themes (experiential, inferential and dispositional), represented in Fig 1. This approach required data-driven coding, deductive thinking and inductive thinking (which are currently used in other approaches to TA [44]) as well as abductive and retroductive thinking (which are advocated in realist methodology, e.g., [45,46]).

As part of the first stage of thematic analysis, the primary author clearly outlined and refined the research aims and objectives (i.e., to identify, explore and understand the features of a sport friendly school in the UK and their impact on the holistic development of student-athletes). In the second stage, the primary author immersed herself in the data, by listening to the audio recordings, undertaking verbatim transcription, reading and re-reading the data to familiarise themselves with the findings and making notes on the initial thoughts and questions. Following familiarization, stage three consisted of applying, developing and reviewing codes (step 3; [42]) to create experiential themes [43]. Experiential themes describe participants' subjective viewpoints and experiences as they are evident in the data [43]. In this study, data on holistic athlete impacts were deductively categorised within four higher-order themes derived from the Holistic Athletic Career model (i.e., academic/vocational, athletic/physical, psychosocial and psychological impacts) with raw data responses inductively coded into more meaningful middle and lower-order sub-themes (e.g., life skills were coded into lower order

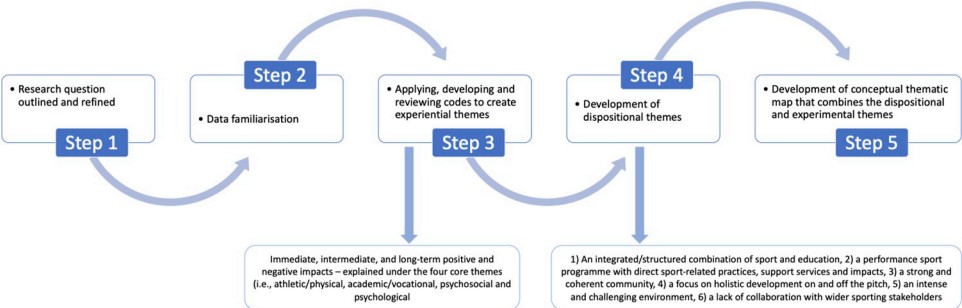

**Fig 1. Thematic analysis process.**

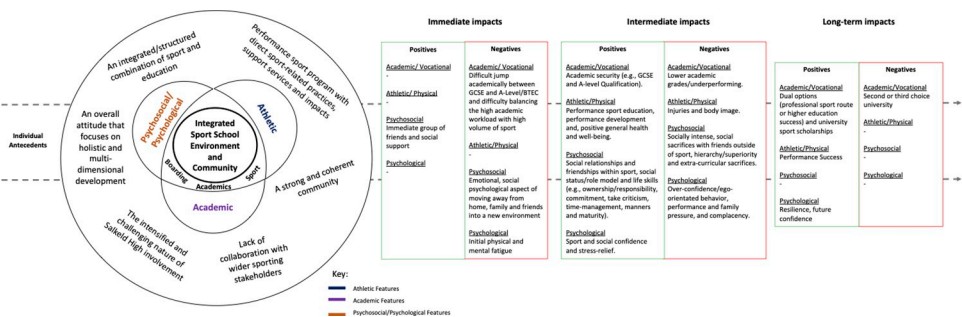

**Fig 2. A conceptual thematic map of "Salkeld High".**

sub-themes such as ownership/responsibility, commitment). Successively, the holistic athlete impacts were categorised under immediate, intermediate and long-term impacts. Immediate impacts referred to impacts experienced by the student-athletes as they transitioned into "Salkeld High". Intermediate denoted to impacts experienced by student-athletes during their time at "Salkeld High" and long-term impacts referred to impacts student-athletes experienced after leaving the "Salkeld High" environment.

Following the development of codes, step 4 entailed developing themes (step 4; [42]) to create dispositional themes [43]. Dispositional themes entail thinking about the mechanisms that have real causal influence on the world [43]. The primary researcher interpreted the data to provide more causal explanations and narrative, outlining how particular features of "Salkeld High" produce the experiences and events evident in the data and codes (e.g., holistic impacts). Finally, within stage five [42], we developed a conceptual thematic map of "Salkeld High" (Fig 2). The diagrams provide a conceptual visualisation, that combines the dispositional themes (e.g., integrated sport school environment and community), with the experiences (experimental themes) of the participants in the research (e.g., immediate, intermediate, and long-term, positive and negative holistic impacts).

## 2.7 Establishing research rigor

In line with our CR approach, we judged the adequacy, plausibility and utility of our explanations as the main criteria to judge the study against [47]. Following the recommendations of Ronkainen & Wiltshire [47], we drew upon Maxwell's [38,48] validity criteria, to establish confidence in the finding's adequacy, plausibility and utility. Adequacy was ensured through recording and precise transcription of all interviews and focus groups, with transcripts thoroughly checked for accuracy against these recordings. Secondly, we foregrounded the voice of the participants and provide quotations throughout the results to provide evidence of empirical justification. In terms of plausibility, frequent peer-debriefing and critical reflections sessions occurred between the researcher and her supervisory team, where the first author's initial interpretations and proposed explanations were reviewed, discussed and challenged [49]. These discussions led to regrouping of subthemes into higher order themes (e.g., sporting and wider community under one main theme—integrated environment and community), and moving from presenting features largely descriptively (listing themes) to a more narrative explanation that captures how these features produce impacts. Furthermore, protocols were put in place to ensure honesty in informants when contributing data (e.g., right to withdraw and reiteration of confidentiality). Finally, regarding utility, we provided practical recommendations and interpretations to guide practical actions in the real-world.

## 3 Findings

The findings are presented and summarised as a conceptual thematic map of "Salkeld High" (Fig 2). The figure describes how individual student-athletes with unique antecedents (personality, family and social background) interacted with the features of "Salkeld High" to produce multiple immediate, intermediate, and long-term positive and negative impacts–explained under the four core themes (i.e., athletic/physical, academic/vocational, psychosocial and psychological; [11]).

"Salkeld High" was viewed as an integrated school environment for sport, academics, and boarding, where educational classrooms, sports facilities and accommodation buildings were all in one location. Within this, "Salkeld High" had numerous athletic (e.g., technical/tactical training, high-quality facilities, high volume/frequency of training), academic (e.g., extra tutoring, dedicated study hours, high academic workload), and psychosocial/psychological (e.g., pastoral services and support network) features. Student-athletes transitioned into "Salkeld High" and interacted with these multiple features, resulting in several immediate, intermediate and long-term positive and negative impacts (see Fig 2). Furthermore, participants described transition over time and how successively, earlier impacts may lead to future long-term positive and negative development impacts.

Below, we now describe in further detail how this integrated sports-friendly school system works, through providing this narrative we formulate an initial programme theory of how the "Salkeld High" environmental features produced impacts. It is worth noting however, that due to the complexity of the interaction of person and environment, that impacts are highlighted through the results, but are not necessarily only linked to the feature discussed. A full list of the potential impacts can be seen in Fig 2.

Overall, "Salkeld High" was viewed as an integrated sport school environment and community, where student-athletes experience:

3.1) An integrated/structured combination of sport and education

3.2) A performance sport programme with direct sport-related practices, support services and impacts

3.3) A strong and coherent community

3.4) A focus on holistic development on and off the pitch

3.5) An intense and challenging environment

3.6) A lack of collaboration with wider sporting stakeholders

### 3.1 An integrated/Structured combination of sport and education

"Salkeld High" was viewed by both student-athletes and staff as an environment that allowed student-athletes to pursue their sporting endeavours alongside their education. In order to help the student-athletes balance their academic and sport workload, "Salkeld High" provided structure for their everyday activities. All of the performance sports students-athletes were enrolled within the standard public-school faculty, where the structure of the curriculum allowed them to combine their sporting programme with a range of educational options in General Certificate of Secondary Education (GCSE), Advanced level qualifications (A-level), and Business and Technology Education Council (BTEC) courses. A streamlined, accommodating curriculum allowed "Salkeld High" to integrate sports training within school hours, with appropriate on-site training facilities and support services. All lessons and sports training were timetabled, providing student-athletes with a structured programme. Athlete 14

expressed how having sport integrated within their timetable freed up more time after school to do other activities, including academic work; *"because we had our training during the school (day), I have the whole of after school to work on my academics and then make sure that I am on top of them as well."* Furthermore, having set dedicated study hours (e.g., timetabled study periods within the curriculum and boarding hours) was emphasised by some student-athletes to provide dedicated time they needed to focus on academic work to ensure they achieved their academic potential.

> *"I think the timetable helps . . . especially having study periods, it gives you just time, to make sure everything is going alright, and you are doing everything you need to. Just making sure we are keeping on top of the work, and with A-level as well, making sure we are revising and getting as good grades as we can."* [Athlete 12]

Further, some student-athletes commented on how having sport timetabled into their day gave them a break from academics, allowing them to regenerate and recharge, as well as a release from other life-stressors, as exemplified by athlete 16, *"in a public school normally, it's just lessons and not really any sport. So, your stress may build up. Where here, you can take out some of your stress with sport because you're realising those endorphins."*

However, due to this highly structured programme (where athletes didn't have to organise their own time), complacency was highlighted as a potential negative impact.

## 3.2 A performance sport programme with direct sport-related practices, support services and impacts

Overall, "Salkeld High" had a performance sports programme embedded within the school. The performance sport programme offered the student-athletes similar support as professional academy programmes (e.g., regular training, S&C, video analysis, nutrition, physiotherapy), alongside education, as highlighted by staff 3: *"You know, being able to do what most professional academy sites do, training every day and doing their S&C, getting the nutrition, the physio. Whilst being at school and still doing the other subjects, it's kind of. . .best of both really."* As part of this programme, student-athletes had access to professional, high-quality facilities (e.g., fully equipped gym, pool, sport science lab, ice baths, etc.), demonstrated by Athlete 18, *"obviously, the facilities here are better than most places in general."* Moreover, embedded within "Salkeld High's" performance programme is a performance sport pathway, where athletes are selected from as young as year 7. As they progress up the pathway (from year 7–13) the training intensity (e.g., number of S&C and training sessions) and access to the support services (e.g., extra tutoring) increased.

In addition, the staff and student-athletes felt that not many other schools could offer the volume/frequency of training highlighted at "Salkeld High", giving the student-athletes more opportunities to improve their physical, technical and tactical skill-based performance and subsequent sporting confidence. *"They're training every day. There are not many schools that can offer that. So, the opportunity they've got to develop is huge compared to other schools or colleges"* [Staff 3]. Furthermore, student-athletes experienced a high standard of competition, including competing against professional academies, which over-time provided more challenging situations to practice and develop key skills.

Staff highlighted how the performance programme was deemed effective, as they recounted that "Salkeld High" produced county, regional, international, and academy/franchise athletes; *"we've had a number of kids who've come to the school who have played representative county [sport], some have played for the North 15s, the North 17s, and we've had two who have played*

*England under 19s."* [Staff 5]. Additionally, student-athletes in the long-term were reported to have gained scholarships to the United States of America (USA) and played for top UK sport universities (e.g., Loughborough University). Finally, "Salkeld High" was seen to provide athletes who were released from academies or franchises with an alternative route and second chance where they were able to carry on training and performing at a high level, providing an opportunity to continue pursuing a professional route.

### 3.3 A strong and coherent community

Overall, "Salkeld High" was perceived to have a strong and coherent community with 1) staff support, 2) community of peers and 3) student-athletes with status and roles in the wider school community.

**3.3.1 Staff support.**   "Salkeld High" was seen as an environment whereby staff were aligned to each support service (i.e., athletic, academic, psychosocial/psychological), valued and endorsed education, providing a streamlined, integrated approach (i.e., who talked to each other and worked together). It was portrayed by both student-athletes and staff that at "Salkeld High", developing the student-athletes in their sport was important, but the 'student' of 'student-athlete' came first. Staff 6 stated, *"But I think first and foremost, we always say this to the students that come here, you know, you're going to leave here with the best exam results possible. That's, that's the driving force behind it, but alongside that, we can get you better at your individual sport as well,"* which was further supported by athlete 18 who highlighted that 'academics always came first.' Over-time gaining academic qualifications was said to have helped provide student-athletes with academic security, maintain perspective and long-term guarantee opportunities for higher education and integration into the workplace at the end of their sports career.

"Salkeld High" had integrated academic support from coaches, house-parents, teachers, and a learning mentor who worked together to mentor and monitor the student-athletes' academic work, helping many of them achieve good academic grades (although not for everyone, as exemplified later). For example, athlete 18 expressed how the coach monitored the student-athlete's grades and checked in regularly with their teacher:

*"Coach is always trying check up on our teachers, monitoring us, and if we are not working hard enough in our lessons, he'll know before you know. Like, if you are not going to get this, then you are not going to play; you're going to go and do work."* [Athlete 18]

Continuing this academic support, teachers provided extra tutoring sessions at lunchtime or after school to support student-athletes academically. *"The support is always there if you ask for it. That is especially true for the academic side. If you are struggling with anything, you can send a message to a teacher and [. . .], they will at least find someone else to help you if they can't help you themselves."* [Athlete 14]

Additional to the academic support received, "Salkeld High" had a dedicated pastoral care team and house-parents who worked to support the welfare and behavioural needs of all students, as exemplified by staff 3: *"pastoral support and the academic school support is, I feel, from my experience of working in different schools, it's probably one of the best. We've got dedicated staff who are there to support. And they are experts in their field."*

Finally, "Salkeld High" was said to have high quality and qualified coaches and a performance sport support staff (e.g., nutrition, S&C and physio support). This high-quality coaching support was perceived by both student-athletes and staff to provide numerous benefits: firstly, foundational knowledge of what it takes to be a professional athlete, aiding their

performance sports education, and secondly, expert coaching and support to further enhance their sporting development. Here staff 3 demonstrated the benefits of his professional experience, "*working with the under 18 students to kind of use my knowledge and experience from the professional games to really push them forward and really demand high standards from them based on my previous professional experience.*" Additionally, the high quality and qualified coaches provided social capital via external contacts and connections with academies, clubs, and universities (USA and UK), which long-term could help some student-athletes link up with professional sports teams and potential scholarship opportunities (e.g., to US university), as emphasised by athlete 16: "*So, our coach generally, has a lot of links, so puts us out there a lot. So therefore, we have more chance of being seen.*"

**3.3.2 Community of peers.**　Overall, at "Salkeld High" there was a community of peers that offered friendship, social skill development, motivation, social support, a sense of belonging and challenging competition in training. At "Salkeld High", the student-athletes were part of a specific sports team and a sporting community of eight performance sports. Many student-athletes perceived being part of this sporting community to support the immediate transition into "Salkeld High", making friends and offering social support.

> "*I was going to say coming into a team environment actually helped me. But going into a team environment and already having like, knowing that they kind of. . . had to be like my brothers on court as well, so then obviously we were going to make good bonds off court, and that really helped. So, I was able to get to know the team quite well and stuff. So, it helped me socially having them I could like speak too, and off that I have made a lot more new friends through my lessons, and my frees and everything.*" [Athlete 17]

Furthermore, due to the integrated school environment, the student-athletes spent a large amount of time together, interacting in class, training (up to seven times a week), competitions (e.g., matches), living (e.g., boarding) and leisure. These factors helped many student-athletes develop friendships/relationships over time.

> "*That camaraderie that there is between the girls, in particular, some of the boys, I'm sure as well, but like. . . yeah, and developing that relationship with teammates and working together on something I know, obviously matches haven't happened, but you know, that team bonding, that is, the seeds are there, yeah, five times a week.*" [Athlete 3]

"Salkeld High" was also described as attracting a big pool of talented student-athletes providing high-quality training partners/teammates. Training partners/teammates were described as influential mentors, and pivotal in providing a high-quality training and learning environment where student-athletes pushed their peers to be better and consequently develop from one another. Athlete 17 explains:

> "*Being pushed by everyone, especially not being advanced, as some of these guys, playing GB, and stuff like that, I have improved a lot. Learning from other players. Like someone like A19 would normally try and help me on court, if I am doing something wrong. Be like you can move there or do this. And he wouldn't say it in a bad way; he would say it in a helpful way. So, it has developed me a lot, and my IQ.*" [Athlete 17]

Lastly, student-athletes interacted with adults (coaches, teachers, house-parents), including many different personalities and cultures (increased sociocultural diversity), which provided

opportunities to develop adult relationships and learn to interact with different people from different backgrounds.

> *"I think the benefit of here as well is that people are coming from different areas—so different cultures. Different schools, when they come in the senior school, and they learn to tolerate people's differences. In terms of not only culturally, but just personality.*
>
> *And I try to drill into my senior girls that you've got to accept that everybody's different on this team."* [Staff 2]

Furthermore, student-athletes has opportunities to engage in public speaking (e.g., assemblies), learnt to work with fellow student-athletes and communicate well within a team, as attributed by athlete 14, *"I think the teamwork that we have, especially with the rugby team, that we always push each other, and we always look for the best in each other."*

As a result, the development of social (e.g., communication, presentation skills) and team skills, were often highlighted as positive impacts of attending "Salkeld High", as exemplified by athlete 4: *"So, I think you get like the communication skills from your coaches. And like being able to build a relationship with an adult that's not your parents."* However, spending large amounts of time together and being in an integrated school environment, was highlighted by some student-athletes as being socially intense and social comparisons. For example, body image was highlighted as a potential negative impact among girls at "Salkeld High". *"It definitely exists particularly among the girls because they, definitely the ones that I know about from last year, they very much compare themselves to other people. So, whilst they're working hard in the gym, they're very conscious of how they look,"* [Staff 1].

**3.3.3 Student-athlete's status and roles in the wider school community.** Student-athletes were also part of a wider school community including non-sport students, academic staff and other staff (e.g., house-parents). Many student-athletes highlighted the status and recognition that immediately came with being a sport performance programme student-athlete within the wider community, as highlighted by athlete 16: *"when you are just walking around, everyone just knows who you are, knows that you are a [sport] player, they'll be like, oh that is the new scholar, and everyone looks up to you and want to be like you."* As a result, many student-athletes were well known and respected. This appeared to help student-athletes' transition into "Salkeld High", develop confidence and make friends.

In addition to the status and recognition of being part of the performance programme (with different age groups), many student-athletes were also role models to their teammates and younger athletes. This appeared to help student-athletes' develop good behaviours due to the role model status, as stated by athlete 16: *"Like, to the little kids you are a role model, and they want to look up to you, so you're setting a good image."* Likewise, the role model status was stated as helping student-athletes' take responsibility and develop leadership skills, as illustrated by staff 3, *"you're the role model. So, we might be the role model for the senior students, but the senior students are the role models for the younger ones. So good leadership skills, set an example, punctuality, things like that."*

However, some athletes also highlighted that in the past, this sense of status had resulted in some older student-athletes thinking they were superior and better than others, and as a consequence, did not buy into the same culture and commitment:

> *"I think also sometimes, some people like last year there was the sixth form and they thought they were older, and they can tell us what to do. This year it's not bad . . . but sometimes I feel like there's like some people that feel like they don't have to do what everyone else asked to do or they can tell everyone else what, but they don't have to do it."* [Athlete 3]

### 3.4 A focus on holistic development on and off the pitch

The staff at "Salkeld High" were seen to provide a holistic approach and intent to their coaching and teaching, and provided an environment where they encouraged the all-round development of the student-athlete, both on- and off the court/field. Here, one athlete explains how they perceived their coach to provide a holistic coaching approach:

> *"What I like the most about coach is not only is he a good [sport] coach, he prepares us for life, he helps us develop as young men, not just as players. He definitely pays as much attention to the off-court things as he does the on-court things, he's always talking to us about our attitude, our mindset, our school, our grades."* [Athlete 18]

The staff shared how they facilitated an environment where over-time many student-athletes built essential life skills, including ownership/responsibility, commitment, the ability to take criticism, time-management, manners and maturity into young men and women. Staff 1 highlighted how he had encouraged some student-athletes to take ownership: *"I think leadership, ownership, I think like, I'm very big on them, like taking ownership and stuff. And like getting them, encouraging them to take over like the warm-ups and get out there."*

Moreover, the performance sport environment at "Salkeld High" incorporated a multi-dimensional training programme, incorporating multiple training modes including technical/tactical, video analysis, psychology, and S&C sessions as part of its programme. *"I've got a programme in place whereby I'm looking at not just the technical aspects of cricket, but I'm looking at tactical, their tactical awareness, their game understanding, and the mental side of the game, and obviously. . .the physical side, the strength and conditioning side,"* [Staff 5]. As a result, as student-athletes progressed through "Salkeld High" they were seen to develop their all-around athletic and sporting performance including, technical/tactical, psychological and physical development. For example, athlete 14 suggested how they developed the technical areas of his/ her sport, helping their overall performance development:

> "*And I think this year, I'm getting to see more of the technical parts, like, okay, if he going there, I have to be there, and if he is going in, I have to go clear out. Which is something that I didn't kind of understand last year, and it has really improved my game being part of the programme.*" [athlete 14].

### 3.5 An intense and challenging environment

The previous themes describe how "Salkeld High" was viewed as an integrated school environment and community, which seemed to provide structured balance for sport and education, a performance sport programme, a strong support community and a holistic focus. Nonetheless, transitioning into a sports-friendly school and balancing academics, sport and social life over-time were also highlighted as potentially intense and providing challenges for student-athletes.

Student-athletes described that moving into a new environment (boarding) and transitioning into "Salkeld High" immediately posed many potential challenges. Such challenges included; the emotional, social psychological aspect of moving away from home, family and friends into a new environment, the difficult jump academically between GCSE and A-Levels/ BTEC and, balancing the demanding academic schedule and high volume/frequency of sport. This initially led to many student-athletes feeling physically and mentally fatigued, however, over-time, student-athletes learnt to balance these demands and were seen to develop many life skills, including independence, efficiency, organisation, discipline, and managing their

time and multiple demands more effectively. *"Definitely, skills that you cannot just use in school life, that you can take, like time management, dealing with pressure. I used to be late to everything. Where we this, I have been a bit changed, in a good positive way."* [Athlete 8]. However, some student-athletes struggled to balance the intensified nature of academics and sport workload. In some cases, becoming too sport-focused, negatively impacting their short- and long-term academic success, with some obtaining second or third choice universities.

> *"They've maybe underachieved in their first year, playing catch up a little bit, probably didn't get the grades that they should have got. Maybe didn't get the university that they were hoping for, so they've had to settle for like a, you know, a second or third choice university because they couldn't quite handle that transition."* [Staff 3]

Furthermore, some student-athletes admitted to having to make extra-curricular and social sacrifices with friends outside of sport.

> *"It's sort of pushing you away from like my friends that don't do sport because a lot of the time when I'm doing sport, they're in like their own little friend group and that's sort of pushed away from that side of the community. Because although like creating a community in sports is good, it could also be a negative because it can create a divide between those who do sport and those who don't so, I think that's like a bit of a negative."* [Athlete 3]

The intensified nature of "Salkeld High" was also described as an environment where student-athletes had to deal with performance pressure on a regular basis (i.e., competition for places within the team, high-level fixtures, scholarship and previous performance success). Many student-athletes developed and learnt skills to deal with this pressure positively, developing long-term resilience, as exemplified by athlete 1, *"it definitely helped me like mentally. But just sort of like resilience and so you when you do fitness, it's sort of like a lot of its mental and stuff like that so it's definitely also made me like realise more about myself and what I can do"*. However, staff did worry that the pressure could have a negative impact on the student-athletes if not managed or controlled.

> *"Some players do feel the pressure of being on a higher scholarship. And the fact that they need to perform to that. A bit of pressure, and I also feel that. . .particularly for my sport, there is a pressure for the team to perform because of the success that we've had previously. And I think for some of the girls, that's a big pressure."* [Staff 2]

This worry was further supported by some student-athletes, as stated by athlete 18: *"I definitely feel that pressure in terms of being able to balancing both side of things, because obviously you want to pursue [sport], but you can't let education drop. Like that pressure for me personally, I didn't find that positive."* Additionally, student-athletes also experienced pressure from families to achieve well in both academic and in sport.

Finally, injuries were also highlighted as a potential negative impact of "Salkeld High", as conveyed by athlete 12, *"yeah, a lot of the lads, are getting. . .pull up on, like injuries, like little injuries like calf or groin or whatever."* Due to the stresses of an injury, this could have had an in-direct negative impact upon academics and the student-athletes psychological health. Here, athlete 15 expressed the negative impact of injuries on academics: *"Injuries, possibility of injuries. Which then affects your academics, and as A13 said, you can get kicked off the programme if your academics aren't good enough. It's kind of a dangerous side, especially playing [sport]."*

### 3.6 A lack of collaboration with wider sporting stakeholders

Lastly, although "Salkeld High" is seen to be an integrated school environment and community the interviews highlighted that it is an isolated system that could better integrate with the wider sporting structures (e.g., representative squads, community sport). Therefore, although the high volume/frequency of training gives the student-athletes more opportunities to play and develop in their sport (as highlighted above), the majority of student-athletes also trained and competed externally to "Salkeld High" (i.e., academies, club, regional and internationally), resulting in a further increase in workload. A lack of workload management between "Salkeld High" and external involvement was stated as causing a challenge for many student-athletes, resulting in a lack of rest and recovery, as exemplified by athlete 12, *"we got obviously the weekend to sort of relax. But if you have games on the weekend, with clubs, it will be a bit difficult."*

## 4 Discussion

This study identified and explored the features of a sports-friendly school in the UK and the holistic student-athlete impacts associated with involvement from the perspective of student-athletes, coaches and teachers. This study is the first to assess the features and impacts of a sports-friendly school across all four areas of holistic athletic development: educational/vocational, physical/athletic, psychosocial and psychological. At "Salkeld High", student-athletes were immersed in an integrated school system of athletic (e.g., high volume/frequency of training), academic (e.g., extra-tutoring), and psychosocial/psychological (e.g., pastoral services) features resulting in multiple positive and negative, immediate, intermediate, and long-term impacts. These impacts are summarised in Fig 2. However, it is worth noting that although there were many similarities, student-athletes interacted with the features of "Salkeld High" differently and thus experienced different impacts. Nonetheless, the student-athletes at "Salkeld High" tended to get a well-rounded, balanced experience of sport, education and psychological/psychosocial development. The balanced experience was supported through a school environment and community that offered 1) an integrated/structured combination of sport and education, 2) a performance sport programme with direct sport-related practices and support services, 3) a strong and coherent community culture and 4) a focus on holistic development on and off the pitch. However, the intensified and challenging nature of the "Salkeld High" involvement (e.g., transition, high workload and pressure) did present some negative impacts that practitioners working in this and similar environments should be aware of. Furthermore, although "Salkeld High" was seen as an integrated environment within the school, it could do at better collaborating with the wider sporting structures.

The student-athletes at "Salkeld High" transitioned into a new environment, immediately experiencing several challenges (e.g., moving away from friends and family) and changes (e.g., intense level of training, academic and boarding). The challenges and changes experienced by student-athletes transitioning into "Salkeld High" are supported by previous research [8,23,50,51]. For example, Finnish sport school student-athletes perceived their sport school routines to be significantly more intense than their previous school schedules [51]. As a result, student-athletes need support to help them prepare for and cope with the challenges and changes of moving into a sports-friendly school environment. Such interventions may include high-quality social support during the transition, meetings between incoming and current/former sports-friendly school students, encouraging parents to prepare their kids for independent living, and workshops in which experts help the new students-athletes with diverse practical issues, such as nutrition, injury preventions, and housekeeping activities to increase their life skills [51,52].

Despite the challenges, many athletes felt their experience over time was well-balanced. One key element was student-athletes' ability to focus on academic development. At "Salkeld High", education was valued by the coaches, teachers and other staff members (e.g., house-parents). This result is similar to Knight, Harwood and Sellars [53], who highlighted the importance of an athlete's support network consistently reinforcing the importance of education and the value of maintaining a dual career. Ensuring the support staff are on the same page and everyone's expectations are aligned, eases tensions within the group and prevents the student-athletes from feeling conflicted [53]. Mentorship, monitoring, extra tutoring and a structured academic curriculum were some of the other academic support services provided at "Salkeld High" which are consistent with previous sport school literature [14,17,23,54,55]. Indeed, personal support in the form of mentors, tutors and personal learning support systems has been recognised as an essential feature for encouraging academic success [56]. Finally, as well as having staff that valued and endorsed education and provided academic support, "Salkeld High" was also perceived to have aligned staff in multiple support areas (i.e., academic, athletic, and psychological/ psychosocial) who through frequent open communication, worked together to support and care for the student-athletes athletic, physical, academic, welfare and behavioural needs. Such a finding is consistent with previous studies [10,53] that have highlighted those integrated efforts as critical features of successful athletic talent development environments. As such, to optimise the holistic development of student-athletes and particular to remove barriers/challenges of being involved in a sports-friendly school, individuals in the support network need to work together rather than in isolation. Furthermore, it seems reasonable that staff value and endorse education and provide extra academic support to help protect student-athletes academic security for future success.

Alongside academic development, the aim of a sports-friendly school is to develop athletic/ physical performance. To date, little was known about what the training at sport schools involved [23]. The current study provided us with an initial insight, demonstrating that through a multi-dimensional training programme, high volume/frequency of training, high-quality coaches and sport support staff student-athletes developed their all-round performance development and sports education and subsequent sporting confidence. This aligns to the views that many trainable factors contribute to sporting success (e.g., [57,58]), that coaches play an essential role in talent development [59–61] and that high-quality sports programmes should incorporate a group of support staff (e.g., S&C coaches, sports psychologists, nutritionists, physiotherapists [62]). As such, it seems plausible that sports-friendly school programmes should be multi-dimensional in nature and employ expert coaches and support sport staff to provide high-quality training programmes and sessions. However, whilst this study demonstrates the value of high-quality coaches and support staff, little is known about how the coaches achieved performance education and development in practice. Furthermore, although high volume/frequency of training was deemed a positive in this study, future research should explore the workload of the sports-friendly school student-athletes objectively and their subsequent correlation with rest, recovery and injury. Finally, in terms of sports success, "Salkeld High" was perceived to produce county, regional and international athletes but future research may explore comparisons with other environments based on the mixed effectiveness reported in the literature [22,23,26,63].

In addition to academic and athletic impacts, social impacts are an important aspect of holistic athletic development. Student-athletes at "Salkeld High" developed friendships and social skills through being part of a team, 'hanging out' with people from a range of backgrounds, having opportunities to speak in public, adapting to many different social situations as well as interacting with adults from a young age. The social opportunity is supported by research that has shown that sports participation provides opportunities for student-athletes to

make friends (e.g., [64]) and develop social life skills (e.g., teamwork, emotional skills, inter-personal communication, social skills, leadership; [65,66]). Further, "Salkeld High" attracted a large pool of talented student-athletes who acted as roles models, facilitating a learning community environment, driving performance development, responsibility and leadership. Role model status is consistent with Henriksen's research (e.g., [55,67,68]), which identified the importance of "having someone to aspire to" and the daily exchange of knowledge and ideas, ultimately driving development and performance standards. As a result, sports-friendly schools should encourage community learning [68–70]. However, negative peer interaction and comparison was highlighted at "Salkeld High" (i.e., social-intensity, hierarchy and superiority), which has been shown to result in negative impacts such as low self-esteem and anger [71,72]. Thus, sports-friendly schools should aim to promote and facilitate positive peer support for effective holistic development [61,73].

The staff were seen to provide a holistic coaching approach that involved sport alongside academic and personal development (e.g., life skills). Although there is limited research exploring the teaching styles/behaviours of coaches within sport schools specifically, previous research supports holistic coaching as a vital factor to student-athlete development, facilitating life skills that help them become better athletes, as well as better people (e.g., [74,75]). Additionally, although some student-athletes struggled with the balance of work and sport, many student-athletes developed life skills through the demanding environment. The requirement to live away from home, the busy schedule and balance of sport and academics required student-athletes to manage themselves effectively, become better at managing multiple demands as well as being disciplined [76]. Life skill development is supported by previous sport school literature, which suggests that sport schools encouraged student-athletes to develop qualities and skills applicable not only in sport but also in other spheres of life [8,55,77]. Previous studies have explored how life skills are transferred from school sport into other life domains. For example, resilience and independence have been considered necessary in coping with the transition to the university level (e.g., [78,79]). However, due to the cross-sectional nature of this study (i.e., limited retrospective perspectives from staff and student-athletes having limited experience to reflect on long-term impacts) it is not clear if and how life skills are transferred from this specific sports school into other life domains.

As highlighted above, combining an athletic career with education is demanding for student-athletes [80]. Some student-athletes at "Salkeld High" found the balance of work and sport a challenging proposition and one that is a concern for most high-performance student-athletes [6]. Therefore, although "Salkeld High" was seen to provide academic support for its student-athletes, some student-athletes had difficulties balancing the intensified nature of academics and sport workload, in some cases becoming 'too sport-focused', impacting negatively on their immediate and long-term academic success. The negative impact of high-performance sport participation [81,82] and the negative consequences of a strong athletic identity (e.g., [83,84]) on student-athletes academic success are supported by previous research. However, numerous sport school studies contradict these negative findings [23] showing that sport schools do not impact negatively on student-athletes academic success [22,26,85,86]. Due to the contradictions within the research, there is a need for greater contextual description in studies linking mechanisms to impacts and further using CR as a way of grounding data in a particular context to allow for greater interpretation. Nevertheless, sports-friendly schools should be aware of the potential risk of student-athletes overly prioritising sport over academics. A strategy to manage this impact may be having a dedicated dual career support team (or person) responsible for coordinating sport and study that helps student-athletes balance their sport and academic workload [87].

In additions to the demands of combing an athletic career with education, injuries were also highlighted as a potential negative impact of "Salkeld High", which is supported by previous sport school literature [23,88,89]. It is impossible to eradicate all injuries from youth sport programmes; however, injury prevention schemes can significantly reduce the frequency and severity of injuries [90,91]. Thompson et al., [23] review provides appropriate recovery and prevention strategies that could be incorporated as part of sports-friendly school programmes. In particular, due to the in-direct stresses of injury upon academics and the student-athletes psychological health, highlighted in this paper, sports-friendly schools need to provide additional social and psychological support to help student-athletes deal with the emotional response to injury [92].

Finally, student-athletes (like those at "Salkeld High") often participate in multiple sports or for various teams within the same sport [93,94]. Therefore, although "Salkeld High" was seen to be a balanced and integrated system within the school, there is a lack of integration with the wider sporting organisations. As a result, juggling the multiple workloads posed challenges for many student-athletes resulting in a lack of rest and recovery. This lack of integration is supported by previous research, which demonstrates the 'tug of war' scenario of various weekly sports commitments, which can result from separate and contrasting athlete-focused training plans and goals [93,94]. Additionally, research demonstrates that student-athletes with higher weekly training loads have higher recovery-stress states than student-athletes with lower weekly loads [95]. As such, sports-friendly schools need to work in tandem with wider sporting organisations in an integrated effort to balance student-athletes training schedules and develop aligned athlete-focused training plans and goals. This will help prevent the unintentional accumulation of fatigue, which may lead to performance decrement, non-functional overreaching, increased likelihood of injury [96,97], and stagnation in physical development.

## 5 Methodological reflections and future research

Whilst this study is the first to assess the features and impacts of a sports-friendly school across all four areas of holistic athletic development, it is also important to be aware of its limitations. Some would argue that due to the first-hand experiences of the primary author, they already had their own preconceived ideas, potentially narrowing the analytic lens of the study. However, to mitigate against this, critical friends were used, there were frequent peer-debriefing and critical reflections sessions between the co-authors, where the first author's initial interpretations and proposed explanations were reviewed, discussed and challenged [49]. Furthermore, student-athletes and staff may have not openly expressed their concerns and opinions in front of a member of the same institution. Nevertheless, the primary author had a strong relationship with the student-athletes and staff impacting their ability (positively) to interact with her, emphasised that all data would be kept strictly confidential, and ensured participants that there were no right or wrong answers and to be open and honest.

Secondly, although there were significant similarities between the student-athletes and staff perspectives on the features of "Salkeld High" and subsequent holistic athlete impacts, some of the impacts and features were largely evidenced from the staff perspective. As such, one would question how much of the evidence is really what happened or what those running the programme believe to happen? This observation supports the need for a more in-depth programme evaluation of "Salkeld High" to develop a more evidence-based understanding of how the programme works.

Due the complexity of the interaction of person and environment there is not always a linear relationship between features and impact. So, while Fig 2 provides a general overview of the features and multiple possible impacts, it is important to note that not every athlete

experienced every potential impact. Instead, impacts across individuals vary and are driven by their individual antecedents and experiences of the features over time. Furthermore, due to the cross-sectional nature of this study, where exposure and outcomes are simultaneously assessed (i.e., asking athletes and staff to reflect now) there is likely a recency effect and athletes are not yet able to communicate future outcomes, making it difficult to determine a temporal relationship [98] and resulting in limited examples of long-term impacts at that time point. Using one single measurement opposes the nature of 'transition' as a process and the dynamic nature of sport-friendly school environments, which calls for a longitudinal approach to investigate student-athletes development or changes over time. Longitudinal data would be incredibly valuable in establishing causal explanations [99]. Furthermore, upon reflection the study interview guide had limited specific attention on the immediate impacts of entering a sports-friendly school programme, which could be explored further in future research.

Finally, future research should provide a sport-by-sport analysis. It is expected that sport schools require an individual approach, tailor-made for each athlete and each sport. Therefore, future research needs to consider the specificity of athlete characteristics/variables (e.g., sex, type of sport, age, development stage within the school, training cycle). Additionally, sports-friendly schools may vary in their resources, organisational structure and aims/objectives, which likely affect whether they provide benefits or contribute to school-age athletes' holistic development [7,100]. Therefore, exploring the holistic impact of sports-friendly schools across different contexts within the UK and countries is warranted to account for the socio-cultural context and local conditions of dual-career programmes. Yet we hope that this exploration can provide a platform for further comparisons with other sports-friendly schools within the UK and different contexts.

## 6 Conclusion & practical implications

Overall, this is the first study to assess the features and impacts of a sports-friendly school across all four areas of holistic athletic development. To optimise the holistic development of student-athletes and in particular to remove the barriers and challenges of being involved in a sports-friendly school, they should (a) have an integrated/structured combination of sport and education (e.g., extra personal academic support and a structured academic curriculum), (b) a performance sport programme with direct sport-related practices, support services and impacts (e.g., multi-dimensional training programme and high-volume/frequency of training), (c) a strong and coherent community (e.g., align staff in academic, athletic, and psychological/psychosocial support areas and encourage community learning and promote and facilitate positive peer support), (d) a focus on holistic development on and off the pitch, (e) be aware of the intensified and challenging nature of sports-friendly school involvement (e.g., provide extra support to help student-athletes prepare for and cope with the challenges and associated changes of entering a sports-friendly school environment), and (f) collaborate with wider sporting stakeholders (e.g., in an integrated effort to balance student-athletes training schedules and develop aligned athlete-focused training plans and goals. Based on the current evidence base we have provided reasonable practical recommendations to aid the design of an appropriate sports-friendly school learning environment that drives positive holistic development (see Table 1). However, further research is needed to gain a more in-depth understanding of how these features are operationalised across different contexts and sports, how they relate to impacts and the longitudinal transition across time to help guide sports-friendly school practitioners and programmes. Furthermore, while this study provides an initial insight into the features of a sports-friendly school in the UK and the holistic student-athlete impacts associated with involvement, there is a need for further contextual description in studies

**Table 1. Summary of practical recommendations.**

| Area | Practical recommendations |
|---|---|
| Transition | • Extra support to help them prepare for and cope with the challenges and changes of moving into a sports-friendly school environment |
| Integration | • Aligned staff in academic, athletic, and psychological/psychosocial support areas, who work together to support and care for the student-athletes holistic development<br>• An integrated/structured timetable for sport and education |
| Academic/ Vocational | • Support network providing a consistent demonstration of the importance of education<br>• A collaborative academic support system (coaches, teachers and other staff members)<br>• Dual career officer<br>• Extra personal academic support e.g., extra-tutoring<br>• A structured academic curriculum |
| Athletic/Physical | • High quality, qualified coaches<br>• Sport science support team (i.e., S&C, physio, nutritionists)<br>• High standard of fixtures and matches<br>• High volume/frequency of training<br>• Multi-dimensional training programmes (incorporating technical, tactical, physical and psychological variables)<br>• Injury prevention schemes (e.g., S&C programmes, monitoring of individual workload, recovery strategies, modifying external training variables to achieve a desired internal response, programme malleability and athlete education, extrinsic factors via the use of protective equipment, and implementation of rules and regulations)<br>• Work in tandem with wider sporting organisations in an integrated effort to balance student-athletes training schedules and develop aligned athlete-focused training plans and goals |
| Psychological/ psychosocially | • Careful management of performance pressures<br>• Holistic approach to coaching<br>• Encourage and advocate for role models/leadership<br>• Encourage community learning<br>• Encourage positive peer support<br>• Additional social and psychological support to help student-athletes deal with the emotional response to injury |

linking mechanisms to impacts. Finally, the individualisation of each student-athletes needs will differ depending upon the sport, the academic pathway, and the individual circumstances [57]. Therefore, flexibility and individualised support is an essential feature for the holistic development of sports-friendly school student-athletes [86].

## Supporting information

**S1 Dataset. Summary dataset.**
(XLSX)

## Author Contributions

**Conceptualization:** Ffion Thompson.

**Data curation:** Ffion Thompson.

**Formal analysis:** Ffion Thompson.

**Investigation:** Ffion Thompson.

**Methodology:** Ffion Thompson.

**Supervision:** Fieke Rongen, Ian Cowburn, Kevin Till.

**Writing – original draft:** Ffion Thompson.

**Writing – review & editing:** Ffion Thompson.

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
