## [Decision Letter · Decision Letter 0]

9 May 2022

PONE-D-22-07870The features and holistic athlete impacts of a UK sports-friendly school: Student-athlete, coach and teacher perspectives.PLOS ONE

Dear Dr. Thompson,

Thank you for submitting your manuscript to PLOS ONE. After careful consideration, we feel that it has merit but does not fully meet PLOS ONE’s publication criteria as it currently stands. Therefore, we invite you to submit a revised version of the manuscript that addresses the points raised during the review process.

We look forward to receiving your revised manuscript.

Kind regards,

Ender Senel, PhD

Academic Editor

PLOS ONE

Journal Requirements:

2. Please change "female” or "male" to "woman” or "man" as appropriate, when used as a noun (see for instance https://apastyle.apa.org/style-grammar-guidelines/bias-free-language/gender).

Reviewers' comments:

Reviewer's Responses to Questions

**Comments to the Author**

1. Is the manuscript technically sound, and do the data support the conclusions?

Reviewer #1: Yes

Reviewer #2: Partly

2. Has the statistical analysis been performed appropriately and rigorously? 

Reviewer #1: N/A

Reviewer #2: N/A

3. Have the authors made all data underlying the findings in their manuscript fully available?

Reviewer #1: Yes

Reviewer #2: No

4. Is the manuscript presented in an intelligible fashion and written in standard English?

Reviewer #1: Yes

Reviewer #2: Yes

5. Review Comments to the Author

Reviewer #1: Review Manuscript PONE-D-22-07870

“The features and holistic athlete impacts of a UK sports-friendly school: Student athlete, coach and teacher perspectives”

Thank you for the opportunity to review this work!

General:

The current study aims to give insights into a Dual Career environment in the UK context by interviewing both student athletes and coaches/teachers about their experience of their immediate, intermediate and long-term development. The manuscript is in general well written and structured, but rather long and in some parts are repetitive.

The current study was an interesting read and advances the literature by providing detailed description of both the benefits and drawbacks of following a dual career at sports-friendly boarding school. Please find some more specific comments below that might further enhance the quality of the manuscripts.

Abstract:

It should be clear that “Salkeld High” is a fictive name or pseudonym from the beginning. I had to read until page 7 to find out about it. I would have it set in quotation marks throughout the text.

Sport friendly school student-athletes. Is school needed here?

Keep the same order of academic/vocational; athletic/physical; psychosocial and psychological in both parts of the abstract.

Introduction:

Generally, well written with relevant references. Gap in literature clearly identified (p. 5). Five gaps are addressed, will all be addressed?

The aim is clear and well-defined. Maybe you could mention Salked High instead of the general SfS expression in the aims.

Methods:

A critical realist approach was adopted, and the authors argue that a qualitative methodology enables the researcher to perceive the impacts of the students involvement in the SfS. I would argue that a case study methodology would fit the research aims better, because Stalked High is clearly your case and the student-athletes and the coaches/teachers and the first authors own experience are the informants about the case. You might want to consider re-formulating your research design.

Context of the study (or case description, l. 155ff): It can be difficult to find the right balance to present the case and then writing in the results section about it. Please see that the information is not redundant (what does the reader need to know before the results to understand the context?).

International people might not be familiar with the abbreviations of GCSE and BTEC.

l. 179. The first author worked (past). Is the work terminated or does it still go on?

Participants: when reading the information about the coaches and teachers, it was not clear who was (only) a coach, or if the always have a double role. Were there any teachers interviewed that have no coaching role?

Data analyses is well described.

l. 280. Figure 1 is clearly an empirical illustration of “Salked High” and should not be termed a programme theory model.

Results:

l. 307 ff. This case description is a repetition of the “Context of the study” from the Method section. Shorten or delete.

l. 335: Heading “Integrated Sport School Environment and Community”. Why not: “Features of Salkeld High” (as in Figure 1?

Just list the 6 features without elaborating here (about nr 4), you do that later in more detail.

l. 352 ff. Again, case description and repetitive. The authors are advised to go through the whole manuscripts to spot passages where the same information is given.

In the results part, focus on the empirical qualitative material (as it is done on p. 18-27).

Discussion:

The discussion is well elaborated and incorporates relevant literature.

Methodological reflection (p. 34). Clearly, the first author has a special role and position through her involvement in the SfS under description. The potential bias is described concerning interpretation of the data. However, it should also be discussed how her position potentially impacted the data collection with student-athletes (power-relation and hierarchy) and coaches/teachers (her peers). Could they really be open and critical when talking to a member of the same institution??

Concerning “transition”: The term has not been introduced before in the Introduction or with references to the broad transition literature. We know from previous case studies (Henriksen et al.) that environments are dynamic and are evolving concerning their staff, members, culture, and their surroundings.

Conclusions: Text and Table 1 (practical implications) is double information. Table 1 is not necessary.

Figure 1: Even though the Figure is quite loaded with information, I like the graphical summary including the development perspective/ outcomes. Should “impact” or “outcomes” be added to immediate/intermediate/long-term?

I hope these comments are helpful. Best of luck in further developing this research!

Reviewer #2: ABSTRACT

- In the Abstract you talk about the 'Impact of sport schools features on the holistic development of student-athletes'... Would you consider reviewing the title and short title to reflect both this aim, and also to reflect the 'case study' approach? This needs to be consistent across the whole study.

- In Abstract, specifcy number of teachers, and coaches. Show the strength of your sample.

- Line 31, Abstract is weak in presenting research Methods.

INTRODUCTION / LITERATURE

- 47-49 Need literature support.

- "Whole Person" - is that a quote? If yes, you need page numbers.

- page 3, 1/3.5 pages does not refer to sports and/or schools! You mention that in line 77 first. A little too late when you consider that the introduction is just 132 lines. Get the focus of the study clear earlier on.

- 82. that/which, instead of 'who permit[s]'.

- 82. Quote is not well integrated in the sentence.

- 82. Lines 82 to 84 you are using ' ' for direct quotes. You should be using " ". Be consistent across.

- 88. Do you consider a new para before 'While exploring...'

- The use of unneccessary acronyms, only makes it harder for the reader. If acronyms are used to reduce word count, then perhaps you need to review your writing. Otherwise, the acronyms make it harder for readers to remember what each one of them means.

- 87 ISCED - in full would give more context to the readers.

- in lines 88-93 you move from Literature to Scope, and then in 94 you move back to literature. Perhaps it is better if you reconsider the posiiton of lines 88-93.

- 94-97. It is not fully clear wheter an ASS = ESS and non-ss = sfs. You need to clarify what a non-ss is. Here it very much sounds like an SFS! Unclear consistency between 94-109 and Morris' Definitions (80-86).

- 131. Perhaps sentence needs to become'of an SfS in the UK and their impact on the holistic development of student-athletes, from the perspective of the student-athletes, coaches and teachers. As discussed earlier, you need to be consistent across the whole study on this.

- Paper needs better consistency in (1) Title and scope, (2) use of coaches/teachers Vs Coaches and Teachers.

- Try to reduce repetitions. Certain things are repeated only few lines later.

METHODOLOGY

- Line 136-142 you mention critical analysis but not thematic analysis.

- First time you mention TA is in line 250, and you do not mention it in 'research approach'

-146-153. Focus more on data collection. Should it be moved to reduce repetition?

- 152. ...and to enable an in-depth...

- 158-163 Can be written better.

- 164. Athletics and swimming wouldnt' probably cover tactics. Or would they?

- 179-180 Presents an issue of confidentiality. We know the authoer, so if one does a search for the authoer one would probably discover Salked High's real name. If I am right, I would say that Salked High in reality is Queen Ethelburga’s (QE) College... if this is yours - https://www.leedsbeckett.ac.uk/blogs/carnegie-xchange/2021/04/the-impacts-of-sport-schools-on-holistic-athlete-development-ffion-thompson/

- 195 and 205. Selection Criteria and actual selection don't fully agree. In line 195 you say that participants had to be... the head coach of one of the programmes. In line 205 you say that one was director of sport. This raises questions. Why?

- 206. 'three of them being teachers' needs to be explained better (consistency with line 212 is important).

- Thematic analysis is not supported by any reference, nor it is clear which process of thematic analysis was applied. Clarke and Braun?

- The concept of Thrustworthiness needs to be underlined, especially to clarify how data led, to the findings in a trustworthy manner, and how themes emerged from that same process.

RESULTS

- I would rather use the term 'FINDINGS' instead of 'RESULTS'. But feel free to decide.

- If the last figure on the last page is Figure 1, it needs a clear caption with numbering.

-320-321. "So I think..." Sentence is not conclusive.

- Ex 318-321. Participants' direct quotes need to be better integrated.

-321-325. It is an important deep analysis of the implications of the model. Set it in a more 'visual' manner, perhaps in a section on its own?

- 326-327. Better in Limitations?

DISCUSSION & CONCLUSION

816-818. Peraps better to underline in Methodology?

IMPORTANT POINTS

- This study needs to clearly indicate the 'Themes', 'Sub Themes, and categories emerging from the data analysis. There seems to be limited coherence in the process of data analysis, presentation of 'results' and 'conclusion'.

- For a Thematic Analysis Approach, it would be better to use a categorisation chart to show how the data led to the themes, sub themes and categories emerging from the data.

- Coherence across the subtitles in the 'results' section and the conclusion / figure 1 may aid in having further clarity as to what the main findings are. Hence, perhaps coherence between the emerging themes (sub themes, categories) and the sections within the results' section would assist in the requested coherence.

- Table 1 is an important figure, but seems to be skipping an important step, that of the 'clarity' of the emerging themes.

- It is not exactly clear what the figure on the last page is presenting - it has no caption to support its position either.

6. PLOS authors have the option to publish the peer review history of their article (what does this mean?). If published, this will include your full peer review and any attached files.

Reviewer #1: **Yes: **Andreas Küttel

Reviewer #2: **Yes: **Renzo Kerr Cumbo

---

## [Author Response · Author response to Decision Letter 0]

22 Jun 2022

Thank you for your thorough review of our manuscript and we appreciate the time taken to conduct such a review. We hope you are satisfied with the changes made to the paper and the points addressed.

---

## [Decision Letter · Decision Letter 1]

21 Sep 2022

PONE-D-22-07870R1A case study of the features and holistic athlete impacts of a UK sports-friendly school: Student-athlete, coach and teacher perspectives.PLOS ONE

Dear Dr. Thompson,

Thank you for submitting your manuscript to PLOS ONE. After careful consideration, we feel that it has merit but does not fully meet PLOS ONE’s publication criteria as it currently stands. Therefore, we invite you to submit a revised version of the manuscript that addresses the points raised during the review process.

We look forward to receiving your revised manuscript.

Kind regards,

Giancarlo Condello, Ph.D.

Academic Editor

PLOS ONE

Journal Requirements:

Reviewers' comments:

Reviewer's Responses to Questions

**Comments to the Author**

1. If the authors have adequately addressed your comments raised in a previous round of review and you feel that this manuscript is now acceptable for publication, you may indicate that here to bypass the “Comments to the Author” section, enter your conflict of interest statement in the “Confidential to Editor” section, and submit your "Accept" recommendation.

Reviewer #1: All comments have been addressed

2. Is the manuscript technically sound, and do the data support the conclusions?

Reviewer #1: Yes

3. Has the statistical analysis been performed appropriately and rigorously? 

Reviewer #1: N/A

4. Have the authors made all data underlying the findings in their manuscript fully available?

Reviewer #1: Yes

5. Is the manuscript presented in an intelligible fashion and written in standard English?

Reviewer #1: Yes

6. Review Comments to the Author

Reviewer #1: “A case study of the features and holistic athlete impacts of a UK sports-friendly school: Student-athlete, coach and teacher perspectives.”

General:

I congratulate the authors for significantly improving the manuscript and for doing a very thorough job by revising the text based on my and the other reviewer’s comments. After reading the new version, I have a few comments or suggestions that the authors may take on board for the next revision.

Abstract:

Well described and improved

Introduction:

Generally, well written with relevant references.

l. 123 “limited research evaluates how features affect athlete impact” is not clear.

l. 127. Kuettel, Christensen, Zysko & Hansen (2020) compared DC environments across Europe and emphasized the different DC mindsets depending on the cultural context.

l. 128-129. The same information is given in line 140-142. And again in 146-147. I believe it is most appropriate to state the aims just before the Method section.

Methods:

Critical realism is better explained and integrated in the project’s description. Participants are well explained. Data analyses and collection are well explained. It could be considered to first explain the collection part and then the analysis strategy and steps.

l. 276 the author immersed herself (singular)

l. 306 “Markers of Quality” heading is not very meaningful. Maybe add something with “research rigor”?

Findings:

The first part of the findings make sense (summarizing the results with Figure 2). However, the part starting from 347 until 367 seems somehow misplaced (is it needed at all?). I also do not recall that resilience was a topic described earlier or later in the discussion.

The part 355-359 belongs more to the discussion part (it is repeated shortly in the methodological reflections)

Heading numbers: Integrated Sport Environment and Community is labelled 3.1. but I could not find any heading 3.2.

Maybe it is more meaningful to label the six themes of your thematic analyses with 3.1 An integrated/structured combination…; 3.2. A performance program…; 3.3. A strong and coherent community; 3.4 ….

Discussion:

The discussion is well elaborated and incorporates relevant literature. Since you present a new conceptual map for exploring DC environments (Figure 2), it would be nice to read more about how your framework is different/better than previous frameworks described in the literature (Henriksen´s ATDE/ESF or DCDE/DC-ESF)

Smith & McGannon reference is double (49 and 98)

I hope these comments are helpful. Best of luck in further developing this research!

7. PLOS authors have the option to publish the peer review history of their article (what does this mean?). If published, this will include your full peer review and any attached files.

Reviewer #1: **Yes: **Andreas Küttel

---

## [Author Response · Author response to Decision Letter 1]

28 Sep 2022

Thank you for your recent review in relation to the submission of the above article for Plos One. 

The review was very thorough and helpful, and has allowed us to consider a number of areas to help us move forward and improve the current paper.

We have addressed the majority of the comments using track changes within the paper and include a response to each of the reviewer’s comments below in relation to the changes that have been made. I hope this is useful in considering our revised submission.

I hope all the below responses and changes to the paper are to the standard expected and hope that the paper is accepted to the journal in the near future.

Yours Sincerely

Ffion Thompson

On Behalf of all co-authors

---

## [Decision Letter · Decision Letter 2]

16 Nov 2022

A case study of the features and holistic athlete impacts of a UK sports-friendly school: Student-athlete, coach and teacher perspectives.

PONE-D-22-07870R2

Dear Dr. Thompson,

We’re pleased to inform you that your manuscript has been judged scientifically suitable for publication and will be formally accepted for publication once it meets all outstanding technical requirements.

Kind regards,

Rabiu Muazu Musa, PhD

Academic Editor

PLOS ONE

Additional Editor Comments (optional):

Reviewers' comments:

Reviewer's Responses to Questions

**Comments to the Author**

1. If the authors have adequately addressed your comments raised in a previous round of review and you feel that this manuscript is now acceptable for publication, you may indicate that here to bypass the “Comments to the Author” section, enter your conflict of interest statement in the “Confidential to Editor” section, and submit your "Accept" recommendation.

Reviewer #1: All comments have been addressed

2. Is the manuscript technically sound, and do the data support the conclusions?

Reviewer #1: Yes

3. Has the statistical analysis been performed appropriately and rigorously? 

Reviewer #1: N/A

4. Have the authors made all data underlying the findings in their manuscript fully available?

Reviewer #1: Yes

5. Is the manuscript presented in an intelligible fashion and written in standard English?

Reviewer #1: Yes

6. Review Comments to the Author

Reviewer #1: (No Response)

7. PLOS authors have the option to publish the peer review history of their article (what does this mean?). If published, this will include your full peer review and any attached files.

Reviewer #1: **Yes: **Andreas Kuettel

---

## [Editor Report · Acceptance letter]

18 Nov 2022

PONE-D-22-07870R2 

A case study of the features and holistic athlete impacts of a UK sports-friendly school: Student-athlete, coach and teacher perspectives. 

Dear Dr. Thompson:

I'm pleased to inform you that your manuscript has been deemed suitable for publication in PLOS ONE. Congratulations! Your manuscript is now with our production department. 

Kind regards, 

on behalf of

Dr. Rabiu Muazu Musa 

Academic Editor

PLOS ONE